# 1 Neodymium isotopes in the ocean model of the Community Earth System

- 2 Model (CESM1.3)
- 3 Sifan Gu<sup>1</sup>, Zhengyu Liu<sup>1</sup>, Alexandra Jahn<sup>2</sup>, Johannes Rempfer<sup>3</sup>, Jiaxu Zhang<sup>1</sup>, Fortunat
- 4 Joos<sup>3</sup>
- 5
- 6 <sup>1</sup>Department of Atmospheric and Oceanic Sciences and Center for Climate Research,
- 7 University of Wisconsin-Madison, Madison, 53705, USA
- 8 <sup>2</sup>Department for Atmospheric and Oceanic Sciences and Institude of Arctic and
- 9 Alpine Research, University of Colorado Boulder, Boulder, 80309, USA
- 10 <sup>3</sup>Climate and Environmental Physics, Physics Institute and Oeschger Center for
- 11 Climate Change Research, University of Bern, Bern, CH-3012, Switzerland
- 12
- 13 Correspondence to: Sifan Gu (sgu28@wisc.edu)
- 14

### 15 Abstract

16 Neodymium (Nd) isotope ratio ( $\varepsilon_{Nd}$ ) is a quasi-conservative water mass 17 tracer and has been used increasingly as paleoclimate proxy to indicate the past evolution of ocean circulation. However, there are many uncertainties in 18 19 interpreting  $\varepsilon_{Nd}$  reconstructions. For the purposes of direct comparison between 20 climate models and proxy reconstructions, we implement Nd isotopes (143Nd and 21 <sup>144</sup>Nd) in the ocean model of the Community Earth System Model (CESM). Two 22 versions of Nd tracers are implemented: one is the "abiotic" Nd in which the particle 23 fields are prescribed as the particle climatology generated by the marine ecosystem 24 module of the CESM under present day forcing; the other is the "biotic" Nd that is 25 coupled with the marine ecosystem module. Under present day climate forcing, our 26 model is able to simulate both Nd concentrations and  $\varepsilon_{Nd}$  in good agreement with 27 available observations. Also, Nd concentration and  $\varepsilon_{Nd}$  in our model show similar 28 sensitivities to the total boundary source and the ratio between particle related Nd and dissolved Nd as in previous modeling study (Rempfer et al., 2011). Therefore, 29 30 our Nd-enabled ocean model provides a promising tool to study past changes in 31 ocean and climate.

32

43

# 33 **1. Introduction**

Radiogenic <sup>143</sup>Nd is produced by the radioactive decay of <sup>147</sup>Sm with decay halflife of 106 billion years (Lugmair, 1974). During magma formation. Nd is more likely 35 36 to enter magma than Sm, therefore, continents have lower Sm/Nd or <sup>143</sup>Nd/<sup>144</sup>Nd 37 compared to mantle (melt residue) and the bulk of earth. The difference of 38 <sup>143</sup>Nd/<sup>144</sup>Nd between continents and the bulk of earth increases with the age of the continent as <sup>143</sup>Nd/<sup>144</sup>Nd in younger continents is more similar to the mantle. 39 40 Therefore, younger (older) continents have higher (lower) <sup>143</sup>Nd/<sup>144</sup>Nd, which is more radiogenic (unradiogenic) (Goldstein and Hemming, 2003). Nd isotopic ratio 41 42 (<sup>143</sup>Nd/<sup>144</sup>Nd) relative to the "bulk earth" value is reported as  $\varepsilon_{Nd}$ :

$$\varepsilon_{Nd} = \left[ \left( \frac{(^{143}Nd/^{144}Nd)_{sample}}{(^{143}Nd/^{144}Nd)_{bulkearth}} \right) - 1 \right] \times 10^4$$

where (143Nd/144Nd)<sub>bulkearth</sub> is 0.512638 (Jacobsen and Wasserburg, 1980). Due to 45 the different ages of continental crust,  $\varepsilon_{Nd}$  in continental crust varies geographically (Albarède and Goldstein, 1992). The general feature consists of the two extremes, 46 with the most unradiogenic values (minimum) in the North Atlantic (-10 to-14), the 47 48 most radiogenic values (maximum) in the Pacific (-3 to -4), and intermediate values in the Indian and Southern Ocean (-7 to -10). Seawater derives its  $\varepsilon_{Nd}$  value mainly 49 50 through weathering and erosion of continental crust (Piepgras et al., 1979). 51 Therefore, different water masses form from different locations have different  $\varepsilon_{Nd}$ 52 values. For example,  $\varepsilon_{Nd}$  of North Atlantic Deep Water (NADW) is around -13.5, whereas  $\varepsilon_{Nd}$  of Antarctic Intermediate Water (AAIW) and Antarctic Bottom Water 53 54 (AABW) is around -8. In the Atlantic,  $\varepsilon_{Nd}$  covaries with salinity (von Blanckenburg, 55 1999) and behaves as quasi-conservative water mass mixing tracer (Goldstein and 56 Hemming, 2003; Piepgras and Wasserburg, 1982).

Unlike quasi-conservative  $\varepsilon_{Nd}$ , Nd concentration shows a nutrient-like behavior 58 as it increases with depth and also along the circulation pathway (Bertram and 59 Elderfield, 1993). The decoupling of  $\varepsilon_{Nd}$  and Nd concentration, or the so-called "Nd 60 paradox", can be explained by reversible scavenging (Bacon and Anderson, 1982; 61 Siddall et al., 2005) in internal Nd cycling (Siddall et al., 2008).

 $\varepsilon_{Nd}$  has been increasingly used in paleoceanographic studies (e.g. Piotrowski et 62 63 al. 2004; Gutjahr et al. 2008; Roberts et al. 2010; Piotrowski et al. 2012) because of 64 its ability to trace different water masses. Also, biological fractionation of Nd 65 isotopes are negligible (Goldstein and Hemming, 2003). However, our knowledge 66 about Nd is limited for a reliable interpretation of  $\varepsilon_{Nd}$  for past ocean changes. For 67 example, interpretation of the Atlantic  $\varepsilon_{Nd}$  reconstructions is based on the assumption of the stable north (NADW) and south (AAIW and AABW)  $\varepsilon_{Nd}$  end-68 69 members. NADW is a mixture of low  $\varepsilon_{Nd}$  water from the Labrador Sea (<-20) and 70 high  $\varepsilon_{Nd}$  water from the Norwegian and Greenland Sea (-7 to -10). Therefore, small changes in deep water formation during the last deglaciation, which is highly 71 72 uncertain (Crocket et al., 2011; Dokken and Jansen, 1999; Labeyrie et al., 1992), will 73 result in large changes in  $\varepsilon_{Nd}$  of NADW (van de Flierdt et al., 2016). In addition, the 74 magnitude and isotopic composition of Nd in sources, which have been suggested to 75 be changing in the past (e.g. : Grousset et al. 1998; Harris and Mix 1999; Amakawa et al. 2000; Lézine et al. 2005; Wolff et al. 2006; Rickli et al. 2010), may also influence 76 77  $\varepsilon_{Nd}$  in seawater (Tachikawa et al., 2003). Therefore, incorporating Nd isotopes into 78 climate models can help to improve our understanding of Nd cycling. Previous 79 modeling efforts of Nd have made much progress (Arsouze et al., 2009; Rempfer et 80 al., 2011; Siddall et al., 2008). Modeling studies also suggest that  $\varepsilon_{Nd}$  end-member 81 changes are relatively small compared with  $\varepsilon_{Nd}$  changes resulted from water mass 82 distribution (Rempfer et al., 2012a) and glacial-deglacial  $\varepsilon_{Nd}$  variations are hard to 83 be obtained by changes in Nd sources alone (Rempfer et al., 2012b).

Currently, many uncertainties and controversies in our understanding of 85 past ocean evolution involve the interpretation of Nd reconstructions (e.g. Huang et al., 2014; Pahnke et al., 2008; Xie et al., 2012). Therefore, it is crucial to incorporate 86 87 Nd isotopes into climate models such that model simulation and proxy record can 88 be compared directly. This direct model-data comparison will help us to better 89 interpret the proxy records and, furthermore, understand past ocean circulation changes. This paper is the documentation of the implementation of neodymium 90 91 isotopes, <sup>143</sup>Nd and <sup>144</sup>Nd, in the ocean model of the Community Earth System Model 92 (CESM) (Hurrell et al., 2013).

The implementation of Nd isotopes in the CESM largely follows Rempfer et 94 al., (2011), which presents the most comprehensive study of Nd modeling to date in 95 the intermediate complexity Bern3D model (Edwards and Marsh, 2005; Müller et al., 96 2006), with a successful simulation of both Nd concentration and  $\varepsilon_{Nd}$  in good 97 agreement with observations. The parameters tuned in Rempfer et al., (2011) are 98 based on the compilation of observations up to September 2011 by Lacan et al., 99 (2012). Nd isotopes are included in the GEOTRACES program (Mawji et al., 2014). 100 van de Flierdt et al., (2016) complied available Nd data up to January 2016, which 101 includes data collected by the GEOTRACES program and additionally approximately 102 1,000 published data points collected outside GEOTRACES. This compilation is more than double the amount of the previous data compilation by Lacan et al., (2012). Our 103 104 study uses the new database to tune model parameters. Also, our Nd module is 105 coupled with the marine ecosystem model of the CESM (eco\_Nd). In addition, we 106 also implement an abiotic Nd (abio\_Nd, similar to Rempfer et al., (2011)). Using a 107 prescribed particle flux field, the abio Nd can be run without the marine ecosystem 108 module and thus has a much-reduced computation cost. Most importantly, the 109 abio\_Nd can be compared with the eco\_Nd to separate the effect of circulation change and biological change on Nd. These two Nd implementations will be added to 110 the code trunk of the current ocean model of the CESM, which will make them 111 112 available to other scientist and will allow them to be maintained as CESM evolves.

This paper serves as a reference for future studies using Nd isotopes in the 114 CESM. We will describe the model and the details of the implementation of the Nd 115 isotopes in Section 2. The experimental design of the test simulations is described in 116 Section 3. The results of the parameter tuning process, the comparison between the 117 simulated Nd concentrations and  $\varepsilon_{Nd}$  with observations, and the model sensitivities 118 to two parameters are discussed in Section 4.

### 120 **2. Model Description**

### 121 2.1 Physical Ocean model

The implementation of Nd is based on the code of CESM, version 1.3. CESM is 123 a state-of-art coupled model and many of the papers describing model component

and analyzing results from CESM can be found in a special collection in Journal of 124 125 Climate (http://journals.ametsoc.org/topic/ccsm4-cesm1). Nd is implemented in the ocean model of the CESM, which is the Parallel Ocean Program version 2 (POP2) 126 127 (Danabasoglu et al., 2012). The experiments in this study are carried out using the 128 fully active and isotope-enabled POP2 coupled to the data atmosphere, land, ice and 129 river runoff under the normal year forcing from CORE-II data (Large and Yeager, 130 2008). The model has a nominal horizontal resolution of 3° and 60 vertical layers, 131 with a 10-m resolution in the upper 200m, increasing to 250m below 3000m. The 132 ocean-alone model at 3° resolution is used due to its low computational cost, allowing us to carry out extensive parameter test simulations. Future applications of 133 the Nd isotopes should use the scientifically validated 1° resolution of the CESM. 134

### 136 **2.2 Biogeochemical component**

The biogeochemical variables used in the Nd isotopes implementation (particle fluxes: CaCO<sub>3</sub>, opal, POC, and dust fluxes) are generated by the marine 138 139 ecosystem model in the CESM (Moore et al., 2013) through the ecosystem driver 140 (Jahn et al., 2015). Simulated annual mean particle (CaCO3, opal, and POC) fluxes 141 leaving the euphotic zone at 105m (Fig. 1,  $a \sim c$ ) show patterns and magnitudes 142 similar to those in satellite observations (Fig. 7.2.5 and 9.2.2 in Sarmiento and 143 Gruber 2006). Surface dust deposition is taken from the ecosystem module, which is prescribed monthly surface dust flux from Luo et al., (2003) (Fig. 1d). The 144 145 remineralization scheme of particle is based on the ballast model of Armstrong et al., (2002). Detailed parameterizations for particle remineralization are documented 146 147 in Moore et al., (2004) with temperature dependent remineralization length scales for POC and opal. 148

### 150 2.3. Nd isotopes implementation

The Nd isotopes (<sup>143</sup>Nd and <sup>144</sup>Nd) are added as optional tracers, which can be turned on at case build time as some other passive tracers (e.g., ideal age, carbon isotopes (Jahn et al., 2015) and water isotopes (Zhang, 2016)). We implement both abio\_Nd and eco\_Nd, the latter of which is coupled with the marine ecosystem model

and therefore requires the ecosystem model to be turned on at the same time. The only difference between abio\_Nd and eco\_Nd is that abio\_Nd uses a set of prescribed annually averaged dust, opal, POC, and CaCO<sub>3</sub> fields that are generated from the ecosystem module offline (Fig. 1), while eco\_Nd uses these fields simultaneously computed from the ecosystem module.

The Nd module is implemented following Rempfer et al., (2011). Nd has 160 161 three sources: river source, dust source, and boundary source. Sedimentation of Nd is the only sink for Nd budget. <sup>143</sup>Nd and <sup>144</sup>Nd are modeled as two separate tracers. 162 163 In addition to <sup>143</sup>Nd and <sup>144</sup>Nd, Nd also has other stable isotopes and the sum of 164 <sup>143</sup>Nd and <sup>144</sup>Nd accounts for 36% of total Nd (Magill et al., 2006). Since we use 36% of the total Nd fluxes as fluxes for <sup>143</sup>Nd and <sup>144</sup>Nd, we need to scale the simulated 165 Nd concentration, which is the sum of <sup>143</sup>Nd and <sup>144</sup>Nd (Eq. (1)), by 1/0.36 when 166 167 compared with observational Nd concentration. Fluxes for <sup>143</sup>Nd and <sup>144</sup>Nd 168 individually are obtained by using a prescribed isotopic ratio (IR, Eq. (2)), which varies for different Nd sources as discussed below. 169

$$Nd = {}^{143}Nd + {}^{144}Nd$$
 (1)

$$IR = {}^{143}Nd/{}^{144}Nd$$
 (2)

### 174 2.3.1 Nd sources

Dust deposition over the ocean surface is one of the Nd sources to the ocean.
The surface dust source, S<sub>dust</sub>(i,j) (g m<sup>-3</sup> s<sup>-1</sup>), is applied to the surface layer of ocean
grid, and can be calculated as:

$$S_{dust}(x,y) = \frac{F_{dust}(x,y) \cdot C_{dust} \cdot \beta_{dust}}{dz_1}$$
(3)

Here, surface dust flux,  $F_{dust}(i,j)$  (g cm<sup>-2</sup> s<sup>-1</sup>), is obtained from the ecosystem module of the CESM (Fig. 1d); global mean Nd concentration in dust is 20 µg/g ( $C_{dust}$ )(Goldstein et al., 1984; Grousset et al., 1988, 1998); 2% ( $\beta_{dust}$ ) of the total Nd in the dust is released into ocean (Greaves et al., 1994); dz<sub>1</sub> (m) is the thickness of the surface layer of the ocean grid. The annual total Nd from dust,  $f_{dust}$ , is 2.1×10<sup>8</sup> g

184 yr<sup>-1</sup>. The individual dust sources for <sup>143</sup>Nd and <sup>144</sup>Nd can be calculated from the
185 prescribed IR field for dust sources following Tachikawa et al., (2003) (Fig. 2c).

190

River runoff also provides Nd to the ocean. The river source of Nd is applied 188 at the surface layer of the ocean. River source,  $S_{river}(i,j)$  (g m<sup>-3</sup> s<sup>-1</sup>), can be obtained 189 from:

$$S_{river}(x,y) = \frac{ROFF(x,y) \cdot C_{river} \cdot (1 - \gamma_{river})}{dz_1}$$
(4)

Here, ROFF(i,j) (kg m<sup>-2</sup> s<sup>-1</sup>) is the river runoff from the coupler of the CESM. The 191 192 simulated global annual river discharge is 41,584 km<sup>3</sup>/yr, similar to the 193 observational estimate of 42,439 km<sup>3</sup>/yr in (Goldstein and Jacobsen, 1987). Nd 194 concentration in river runoff, Criver (g kg<sup>-1</sup>) is extrapolated from the river Nd concentration data (Goldstein and Jacobsen, 1987); we assume 70% ( $\gamma_{river}$ ) of Nd in 195 rivers is removed in estuaries, following Rempfer et al., (2011);  $dz_1$  (m) is the 196 197 thickness of the surface layer of ocean grid. The annual total Nd source from river runoff,  $f_{river}$ , is  $1.3 \times 10^9$  g/yr, which is larger than the reported values of  $5 \times 10^8$  g/yr 198 (Goldstein and Jacobsen, 1987). The difference may be caused by the extrapolation of Nd 199 concentration from the original data from Goldstein and Jacobsen, 1987. Individual river 200 sources for <sup>143</sup>Nd and <sup>144</sup>Nd can be calculated from the prescribed IR field following 201 202 Jeandel et al. (2007) (Fig. 2b).

Weathering of continental crust is another source of Nd. This boundary 205 source of Nd is applied to all continental margin grids above 3,000 m (Fig. 2b). We 206 assume a globally uniform boundary source per unit area (fboundary/Atot), where 207  $f_{boundary}$  (g/yr) is the total boundary source and  $A_{tot}$  (m<sup>2</sup>) is the total area of 208 continental margin. f<sub>boundary</sub> is a tuning parameter, as in Rempfer et al., (2011). 209 Boundary source used in Arsouze et al., (2009) is assumed to be exponentially decreasing with depth but observations from GEOTRACES data suggest no obvious 210 211 depth dependence (van de Flierdt et al., 2016). Boundary source, S<sub>boundary</sub>(x,y,z) can 212 be calculated by Eq. 5, where  $dz_k$  is the thickness of the ocean grid layers. Individual

- boundary source for <sup>143</sup>Nd and <sup>144</sup>Nd can be calculated from the prescribed IR field
- following Jeandel et al. (2007) (Fig. 2 b).

$$S_{boundary}(x, y, z) = \frac{f_{boundary}}{A_{tot}} \cdot \frac{1}{dz}$$
<sup>(5)</sup>

216

### 217 2.3.2 Reversible scavenging and Nd sink

Reversible scavenging is the process of Nd adsorption onto sinking particles 219 (POC, opal, CaCO3, and dust) and desorption during particle dissolution at depth, which transports Nd downwards (Siddall et al., 2008). Total Nd can be separated 220 221 into dissolved Nd phase ( $[Nd]_d$ ) and particle associated Nd phase ( $[Nd]_p$ ) (Eq. (6)). 222 Particle associated Nd can be further separated into Nd associated with different 223 particle types (POC, CaCO<sub>3</sub>, opal, and dust)(Eq. (7)). At the bottom grid, Nd 224 associated with undissolved particles is removed from the ocean through 225 sedimentation, which is the sink for Nd budget.

$$[Nd]_t^j = [Nd]_p^j + [Nd]_d^j$$
 (6)

$$[Nd]_{p}^{j} = [Nd]_{p,POC}^{j} + [Nd]_{p,CaCO_{3}}^{j} + [Nd]_{p,opal}^{j} + [Nd]_{p,dust}^{j}$$
(7)

The ratio of between dissolved [Nd]<sub>d</sub> and [Nd]<sub>p</sub> is given by the "equilibrium
scavenging coefficient", K:

$$K_i^j = \left(\frac{[Nd]_p}{[Nd]_d}\right)^j \cdot \frac{1}{\overline{R_i}},\tag{8}$$

where i refers to different particle types (i = POC, CaCO<sub>3</sub>, opal, and dust), j refers to the different Nd isotopes (j = <sup>143</sup>Nd and <sup>144</sup>Nd), and  $\frac{[Nd]_p}{[Nd]_d}$  here is another tuning parameter.  $\overline{R}_i$  is the ratio between the global average particle concentration ( $\overline{C}_i$ , Table1) and the average density of seawater (1024.5 kg m<sup>-3</sup>). We assume the dissolved Nd and the particle associated Nd are in equilibrium as in other studies (Arsouze et al., 2009; Rempfer et al., 2011; Siddall et al., 2008). Therefore, in each grid cell, the ratio between [Nd]<sub>p</sub> and [Nd]<sub>d</sub> can be obtained from Eq. (9).

$$(\frac{[Nd]_{p,i}(x,y,z)}{[Nd]_d(x,y,z)})^j = K_i^j \cdot R_i(x,y,z)$$
(9)

where  $R_i(x,y,z)$  is the ratio between the particle concentration,  $C_i(x,y,z)$ , and the 241 density of seawater.  $C_i(x,y,z)$  can be calculated from particle fluxes  $F_i(x,y,z)$ , which are provided by the ecosystem module, by applying a settling velocity (w) ( $C_i$  = 242  $F_i/w$ ). We assume a uniform settling velocity of 1000 m yr<sup>-1</sup> for all four kinds of 243 244 particles. This is the velocity of the small particles, which drives the vertical cycling 245 of isotopes (Arsouze et al., 2009; Dutay et al., 2009; Kriest, 2002). Isotopic fractionation between <sup>143</sup>Nd and <sup>144</sup>Nd during the reversible scavenging process is 246 247 neglected as in Rempfer et al., (2011) because of similar molecule mass of <sup>143</sup>Nd and <sup>144</sup>Nd. We, therefore, apply the same  $K_i$  to <sup>143</sup>Nd and <sup>144</sup>Nd. 248

The reversible scavenging process acts as the internal cycling of Nd, which transports Nd from shallow layers to deep layers. This process can be quantified as a source term in the Nd equation

$$S_{rs}^{j}(x,y,z) = \frac{\partial (w \cdot [Nd]_{p}^{j}(x,y,z))}{\partial z},$$
(10)

where w is the settling velocity of particles (1000m yr<sup>-1</sup>) and  $[Nd]_p$  is Nd associated with particles, which can be calculated at every time step using Eq. (6), (7) and (9) (combined as Eq. (11)).

$$[Nd]_{p}^{j} = [Nd]_{t}^{j} \cdot (1 - \frac{1}{1 + K_{POC} \cdot R_{POC} + K_{CaCO_{3}} \cdot R_{CaCO_{3}} + K_{opal} \cdot R_{opal} + K_{dust} \cdot R_{dust})$$
(11)

Therefore, the conservation equation for Nd can be written as

$$\frac{\partial [Nd]_t^j}{\partial t} = S_{dust}^j + S_{river}^j + S_{boundary}^j + S_{rs}^j + T([Nd]_t^j),$$
(12)

such that the Nd concentration change is determined by three source terms in Eq.
(3), (4) and (5), as well as the reversible scavenging term in Eq. (10) and the oceanic
transport term (T).

## 265 3. Experiments

Following Rempfer et al., (2011), our Nd model is tuned with two parameters:  $f_{boundary}$  and  $\frac{[Nd]_p}{[Nd]_d}$ , in the abio\_Nd implementation under present-day climate forcing. The tuning in the abio\_Nd implementation gives us a great computational efficiency because the ecosystem module can be turned off. Yet, the parameters tuned for abio\_Nd should also apply to the eco\_Nd since the particle fields used in the reversible scavenging process for abio\_Nd are the climatology taken from the equilibrium ecosystem module under the same climate forcing.

We have run 99 sets of experiments with different combinations of fboundary 273 and  $\frac{[Nd]_p}{[Nd]_d}$  to search for the optimal set of parameters that can simulate both Nd 274 275 concentration and  $\varepsilon_{Nd}$  most consistent with available observations. These 276 experiments also help us to understand the sensitivity of Nd concentration and  $\varepsilon_{Nd}$ to these two parameters. We have varied  $f_{boundary}$  from 1×10<sup>9</sup> g yr<sup>-1</sup> to 8×10<sup>9</sup> g yr<sup>-1</sup> 277 (more specifically, 1×10<sup>9</sup>, 2×10<sup>9</sup>, 3×10<sup>9</sup>, 4×10<sup>9</sup>, 5×10<sup>9</sup>, 5.5×10<sup>9</sup>, 6×10<sup>9</sup>, 7×10<sup>9</sup>, 8×10<sup>9</sup>) 278 and  $\frac{[Nd]_p}{[Nd]_d}$  from 2×10<sup>-4</sup> to 18×10<sup>-4</sup> (more specifically, 2×10<sup>-4</sup>, 4×10<sup>-4</sup>, 6×10<sup>-4</sup>, 8×10<sup>-4</sup>, 279 9×10<sup>-4</sup>, 10×10<sup>-4</sup>, 11×10<sup>-4</sup>, 12×10<sup>-4</sup>, 14×10<sup>-4</sup>, 16×10<sup>-4</sup>, 18×10<sup>-4</sup>), similar to Rempfer et 280 281 al., (2011). We have not run experiments with  $f_{boundary}$  equals to  $0 \times 10^9$  g yr<sup>-1</sup>, as 282 experiments with low fboundary overall show unrealistic Nd inventory or Nd 283 concentration.

The Nd concentrations (143Nd and 144Nd) are initialized from zero and each 285 experiment is integrated for 4,000 model years (experiments with  $f_{boundary} = 1 \times 10^9$ and  $2 \times 10^9$  are initiated from 3,000 model years of the experiment with  $f_{boundary} =$ 286 287  $3 \times 10^9$  and then integrated for another 1,300 model years each). Nd inventory has 288 reached equilibrium in most of the experiments at the end of the simulation. Those that do not reach equilibrium show unreasonable Nd concentrations or  $\varepsilon_{Nd}$  and drift 289 290 further and further from observation as the model integrates (e.g. cases with  $\frac{[Nd]_p}{[Nd]_d}$  < 6×10<sup>-4</sup>), and therefore are terminated at some point. 291

**4. Results**