# Peer review of "Neodymium isotopes in the ocean model of the Community Earth System"

_Geoscientific Model Development, 2017_

## Referee Comment (RC1) · Anonymous Referee #1 · 7 Apr 2017

Neodymium (Nd) isotopes are useful tracers increasingly used as a paleoclimatic proxy of oceanic circulation evolution. In this paper, the authors have implemented Nd isotopes in the ocean model of the community Earth System Model (CESM). This implementation follows a relatively long series of Nd modeling efforts during the past 15 yearsÂă: Tachikawa et al. (2003), Arsouze et al. (2007, 2008, 2009, 2010), Siddall et al. (2008), Jones et al. (2008), Rempfer et al. (2011, 2012), Sepulchre (2014) or Ayache et al. (2016). All these studies were motivated by the needs of two main scientific communities: geochemical (to better constrain the oceanic cycle of the Nd isotopes and its potential as an oceanic circulation proxy) or paleoclimatic (to understand the past variations of oceanic circulation and comparison with Nd archives). This present paperonly thoroughly reproduces (including the exact same sensitivity tests and diagnostics) with the CESM model, what has beendone in the study by Rempfer

et al. (2011) with the Bern3D model and does not provide any new insight on either of the scientific understanding or technical aspects of what has been done in the previous papers. An interest of such a study could have included an accurate comparison between these new results and Rempfer et al. (2011) (or Arsouze et al. (2009)) in order to identify if some specific drawbacks of one model in some areas were not reproduced by another model, etc... but most of the time the description of the comparison is limited to "similar". GMD papers should propose cutting-edge new parameterizations or enhancements in existing models, motivated by our limits in answering scientific questions. This already existing (first implementation published in 2009 !) implementation of Nd isotopes in the CESM oceanic model is clearly missing a scientific objective, whether motivated by numerical or geochemical questionings.

The main interest of this paper lies in a new modeling tool made available for the CESM users community, and therefore should probably be released as a technical internal note. The article is rather clear and well written. The validation of the simulations, although too descriptive and simple, is done thoroughly and should provide a useful (internal) reference for future publications of Nd isotopes simulations with CESM model. On reading this paper, I do not see significant improvement concerning the understanding of the Nd oceanic cycle, nor the improvment of the Nd cycle modeling, nor the intercomparison of Nd modeling with different oceanic models which could be of interest to the GMD readership. Hence, I cannot recommand this article for publication in GMD.

Please find below some comments below:

The authors consider a $\pm$ 3eNd metric to validate the simulations. However, variations in Nd archives are often within this range. This strong limitation to the validation should be mentioned.

On all the results and figures: it is not clear what variable is shown ? Is it a snapshot of the last month or last year of simulation ? Is it averaged over the last XX years ?

Most a the description of the simulation refers to Rempfer et al. (2011), but Rempfer et al. (2011) originally refers to Arsouze et al. (2009) and Tachikawa et al. (2003). This should be corrected.

l61: Siddall et al. (2008) state that both lateral advection and reversible scavenging are needed

l61: you should probably also mention Boundary Exchange as a possible process to explain the Nd paradox

l129: can you specify the time period simulated ("normal year") and the interannual variability of your model ? I am not sure to understand: do you run the full CESM or do you read oceanic fields offline from a pre-computed simulation ?

l132-134: This paper is a validation paper but it is stated that future simulations (in particular for paleoclimate studies) will be carried out with a different version (higher resolution) of the model. I understand the interest of performing a large range of simulations with a low resolution model to optimize coefficients but at least one simulation (and even BS-XX and PD-XX simulations) at 1° resolution should have been performed. Does the higher resolution model improve the Nd results ? Are the parameters selected relevant at higher resolution ? Are they still optimum ? Is the sensitivity to parameter changes the same ?

l154: you need to tell a little bit more about the particle fieldsÂă: what is exactly the difference between "eco" and "abio" ? I understand that "abio" comes from an output of a previous simulation, but what is this simulation ? Does the "eco" simulation use the same setup for the biogeochemical model as the one used to generated the "abio" fields ? In other words: do differences between those two simulations only reflect online vs offline effect or also a change in the particle concentrations / fluxes ? Do you expect an optimized coefficient change with consequent particles distribution changes (as possibly expected in paleo studies) ?

l397: can't this shift in depth be attributed to a too sluggish AMOC that favors vertical cycling rather than lateral advection ? Actually, the core of NADW visible from eNd data is rather 1500m than 3000m used here. l404-406: it looks that you use the same justification as Rempfer et al. (2011). What could be the drawbacks of the "sources simplifications" ? As you have a higher resolution of the CESM model available, did you test if improving the resolution helps reducing your biases ?

l425: although this sensitivity test has been performed in Rempfer et al. (2011), you should precise what your motivations are for performing such a sensitivity test.

throughout the text: fbounday → fboundary

Fig2: it does seem that you include oceanic ridges in your sources. Wether ridges are sources (in the Pacific ?) or sinks (as we'e thought for a long time), it is very unlikely that values can be -10/-5 eNd.

Fig8: should put a colorbar here

Fig12: if you only look at online vs offline effects here (not sure this is the case), we would rather expect to see variations near the surface, which should be more relevant to look at.

---

## Referee Comment (RC2) · Anonymous Referee #2 · 14 Apr 2017

To directly compare climate models and proxy reconstructions, the authors implemented Nd isotopes in the ocean model of the Community Earth System Model (CESM). Tuning parameters, fboundary (Nd flux from continental margins) and [Nd]p/[Nd]d (Nd concentration ratio between particulate and dissolved phases), were optimized based on the cost function of [Nd] and $\varepsilon$Nd. Since this study provides Nd isotope code to the CESM community, rigorous validation is appreciated. The way to optimize fboundary and [Nd]p/[Nd]d is identical to the previous study by Rempfer et al. (2011) who had used intermediate complexity model Bern 3D model. Considering that the spatial resolution of CESM is higher than Bern 3D and that available seawater Nd concentration and $\varepsilon$Nd data were almost doubled meantime, the authors could provide significant advance about oceanic Nd cycle by numerical modelling. Nonetheless this study just confirmed the findings of the previous study. Indeed, the objective of the

study about the oceanic Nd cycle is too general and originality relative to the previous studies is not clear. The comparison between "abiotic" and "biotic" Nd was proposed as a novel point although only small part of text is attributed to this theme because of very small difference of simulated [Nd] and $\varepsilon$Nd.

This MS has a good potential to contribute to the improvement of our knowledge on oceanic Nd cycle even if the authors did not try to go further (maybe it would be realized by future works). If the authors' objective is to present this paper as a reference for future studies using Nd isotopes in the CESM, more efforts will be required. My major concerns are (1) the way to optimize tuning parameters and to evaluate simulation performance and (2) the assumption of homogeneous Nd flux from margins. Below I develop my suggestions and comments.

(1) Way to optimize tuning parameters and evaluate simulation performance fboundary and [Nd]p/[Nd]d are optimized by cost function J (Figure 3). There is no information about spatial distribution of difference between observation and model simulation except for several selected profiles (Figures 9 and 10). Histograms (Figures 6 and 7) only present the trend in major oceanic basins for four depth layers. The size and spatial distribution of difference between observation and model will provide the information about under and/or over-estimation of source and sink terms. For instance, I would like to see the results of the tuning that is realised separately for different oceanic basins (Atlantic and IndoPacific separately and Southern Ocean as buffer zone to ensure the continuity, for example). The upper layers affected by dust and river water will be treated separately from and the lower layers. With higher spatial resolution and more observational data relative to the previous study, such treatment would be possible. Since lithology and distance from continental margins are different between Atlantic and IndoPacific, it is not surprising that different parameterization lead to better simulation of seawater Nd concentration and $\varepsilon$Nd values. About the evaluation of simulation performance, the authors continued to use a track of vertical sections from Atlantic to Pacific (Figure 2a). Because of large gradient of Nd concentration and $\varepsilon$Nd

[Figure]

from Atlantic to Pacific, moderate amplitude of discrepancy is not visible with this presentation. Basin-scale transect is more appropriate for this study. About the criteria of good agreement ($\pm$ 3 $\varepsilon$-units) should be revised because this size is equivalent or larger than changes in glacial/interglacial intermediate/deepwater $\varepsilon$Nd values.

2) Assumption of homogeneous Nd flux from margins This assumption was already questioned in the study of Rempfer et al. (2011) by the authors themselves ("a globally uniform flux of fbs probably is not valid"). It will be really interesting to tackle this difficult issue because there are some new evidences. The first clue is the partial dissolution of river particle. This potential source had been considered independently from margins before the idea of the boundary source is generally accepted. A recent study on Amazon river mouth demonstrates the dissolution of detrital fraction and Nd release to the ocean (Rousseau et al., 2015). Since river runoff was simulated in CESM, river sediment flux could be quantitatively evaluated by assuming ratio(s) between dissolved and solid phases, a partial dissolution rate and a Nd concentration in solid phase. It is a similar treatment to dust Nd flux. This consideration will contribute to establishing weighted Nd flux from margins. The second clue is Nd release from poorly chemically weathered detrital fraction in relation to the dynamics of cryosphere (Howe et al., 2016). Howe et al. (2016) indicated detrital Nd contribution in the Labrador Sea due to Laurentide ice sheet retreat in the early Holocene. At present, glacier and ice sheet retreat at high latitudes during warm seasons could form Nd flux to the ocean by similar processes. Even if it will be difficult to quantitatively estimate such Nd flux, some sensitivity tests will provide new insight into Nd flux from this source.

Considering a high potential of this work and significant points to be revised, I recommend an overhaul revision and eventual resubmission of the work.

Specific or minor comments Figures 9 and 10: What are the criteria of selection to show the profiles comparing Nd and $\varepsilon$Nd values between observation and simulation?

More recent compilation of seawater Nd and $\varepsilon$Nd as well as Holocene $\varepsilon$Nd values of

Interactive
comment

sedimentary authigenic fraction and biogenic carbonate by Tachikawa et al. (in press) provides hydrography parameters (temperature, salinity, nutrients) that could be useful for data model comparison, for instance with Figure 11.

References Howe, J. N. W., Piotrowski, A. M., and Rennie, V. C. F.: Abyssal origin for the early Holocene pulse of unradiogenic neodymium isotopes in Atlantic seawater, Geology, doi: 10.1130/g38155.1, 2016. 2016.

Rousseau, T. C. C., Sonke, J. E., Chmeleff, J., van Beek, P., Souhaut, M., Boaventura, G., Seyler, P., and Jeandel, C.: Rapid neodymium release to marine waters from lithogenic sediments in the Amazon estuary, Nat Commun, 6, 2015.

Tachikawa, K., Arsouze, T., Bayon, G., Bory, A., Colin, C., Dutay, J.-C., Frank, N., Giraud, X., Gourlan, A. T., Jeandel, C., Lacan, F., Meynadier, L., Montagna, P., Piotrowski, A. M., Plancherel, Y., Pucéat, E., Roy-Barman, M., and Waelbroeck, C.: The large-scale evolution of neodymium isotopic composition in the global modern and Holocene ocean revealed from seawater and archive data, Chem. Geol., doi: http://doi.org/10.1016/j.chemgeo.2017.03.018, in press.

———————————

---

## Author Comment (AC1) · 21 Apr 2017

We thank the reviewer for his/her time for constructive comments.

We are sorry that we have not made it sufficiently clear that the main objective of this paper is a documentation of Neodymium isotopes in the CESM. To emphasize this, we will emphasize this in several additional places in the paper. Indeed, this paper is a follow-up of Jahn et al. 2015, which describes carbon isotope implementation in the CESM. We are implementing different isotopes in the CESM for the purpose of the capability of a direct model-data comparison, which will help the community to better understand past climate changes in terms of better interpretations of different proxy records as well as model validation. We first implement this isotope in the CESM and will be using this module to explore some paleoclimate problems. For example,

we are currently using this tracer module to resolve the controversy of available $\varepsilon$Nd reconstructions in tropical Atlantic (Huang et al., 2014; Pahnke et al., 2008; Xie et al., 2012). Since CESM is a community model and will be used by many users, we were told it is necessary to document Nd implementation in the CESM. Thus, this is a technical paper, which describes and documents a new feature of the CESM. This is why we submit it to GMD. As far as we know, this is the first attempt to implement Nd in the CESM. The reviewer pointed out that the already existing implementation of Nd in the CESM ocean model in 2009, but we cannot find any references regarding this. We have no information on this from our NCAR collaborators either.

In addition to the documentation purpose, we do have some other points to make. First, we follow the methods in Rempfer et al., 2011 since it is the most comprehensive survey of Nd cycle. We use their method, but we are implementing in a different model. Bern3D is an intermediated complexity model, but CESM is a much more sophisticated model. It is not obvious that the two independent models have to produce similar results. In addition, we use a more completed data set to tune model parameters (line 97-104), which yields similar parameters as in Rempfer et al., 2011. A totally different model, as well as double the amount of available observations, give similar results. In some sense, our work is a confirmation of the robustness of Rempfer et al., 2011.

Second, different from Rempfer et al., 2011, we implemented the abiotic Nd module, which uses fixed particle fields as Rempfer et al., 2011, as well as a full biotic Nd module, which is coupled with active marine ecosystem calculating particle fields simultaneously. The abiotic and biotic Nd shows identical results under present forcing because the particle fields used in abiotic Nd module is the climatology of the particle fields in the marine ecosystem. Therefore, in principle, when equilibrium is reached, abiotic and biotic should produce the same climatological Nd fields (there are some seasonal variations, but the comparison showed in the paper are all decadal mean). However, if we use this module to do paleo simulation, in which both circulation and particle fields are changing, these two versions of Nd have the advantage that we can

separate the two effects. But this kind of simulation is beyond the scope of this study and we are working on extensive paleo simulations and will be addressed in ongoing work.

In the following, we have addressed all comments, with the original review quoted.

"The authors consider a +/- 3eNd metric to validate the simulations. However, variations in Nd archives are often within this range. This strong limitation to the validation should be mentioned."

Thanks for pointing this out. Yes, we should mention this limitation in the text. This +/- 3 $\varepsilon$Nd for validating is from Rempfer et al., 2011, we use their measurement as a benchmark, therefore we can compare with their results.

"On all the results and figures: it is not clear what variable is shown ? Is it a snapshot of the last month or last year of simulation ? Is it averaged over the last XX years ?"

We should state this clearly. All the results and figures are based on the latest ten years average (decadal mean) in each experiment.

"Most a the description of the simulation refers to Rempfer et al. (2011), but Rempfer et al. (2011) originally refers to Arsouze et al. (2009) and Tachikawa et al. (2003). This should be corrected."

Thanks for pointing this out. Maybe we should explicitly say that Rempfer et al. 2011 is based one previous works. We also cited Arsouze et al. 2009 and Tachikawa et al. 2003 where similar methods are involved (For example, line 185, line 237, line 245).

"l61: Siddall et al. (2008) state that both lateral advection and reversible scavenging are needed"

In Siddall et al. 2008, they pointed out that the Nd paradox can be explained by the combination of both lateral advection and reversible. We should also include lateral advection here. Any tracer is subjected to ocean transport, but the reversible scavenging

is unique.

"l61: you should probably also mention Boundary Exchange as a possible process to explain the Nd paradox"

Boundary Exchange, which is Nd exchange between water and sediment, is more related to the source-sink for Nd cycle. Nd paradox is the decoupling of Nd and $\varepsilon$Nd. We think reversible scavenging and lateral advection are more important in explaining Nd paradox.

"l129: can you specify the time period simulated ("normal year") and the interannual variability of your model ? I am not sure to understand: do you run the full CESM or do you read oceanic fields offline from a pre-computed simulation ?"

CESM has different components: atmosphere, ocean, land, ice and river. Different components communicate through coupler. These components can be run together as fully coupled and also can be run alone with other components as data. For example, in this study, we run ocean alone experiment (for the purpose of reducing computational cost): ocean is active, but other components are data (data-atmopshere, data-land, data-ice and data-river). Ocean component does not know whether the fields passed through coupler are data or active. Ocean component is active. It is not offline.

CORE dataset (Large and Yeager, 2008) is based on NCEP reanalysis and satellite observation and provides a method to run ocean model without a fully coupled GCM. This method has been widely used in ocean alone simulations. It has two options: one is "normal year forcing", which is repeating seasonal cycle and no interannual variability in this forcing; another is the interannual forcing.

"l132-134: This paper is a validation paper but it is stated that future simulations (in particular for paleoclimate studies) will be carried out with a different version (higher resolution) of the model. I understand the interest of performing a large range of sim-ulations with a low resolution model to optimize coefficients but at least one simu-

lation (and even BS-XX and PD-XX simulations) at 1_ resolution should have been performed. Does the higher resolution model improve the Nd results ? Are the parameters selected relevant at higher resolution ? Are they still optimum ? Is the sensitivity to parameter changes the same ?"

We use the 3 degree ocean as in Jahn et al., 2015. This Nd module will be eventually validated in fully coupled CESM (active atmosphere, land and etc.) at 1 degree resolution along with all other isotopes (e.g. carbon isotope, water isotope) in the future. However, we think the results should be similar. The reason is that our results (optimal parameterization, model sensitivity to parameters) are similar to Rempfer et al. 2011, which is from a totally different model (Bern3d, intermediate complexity) at much lower resolution.

"l154: you need to tell a little bit more about the particle fieldsÂËŸ a: what is exactly the difference between "eco" and "abio" ?  I understand that "abio" comes from an output of a previous simulation, but what is this simulation ? Does the "eco" simulation use the same setup for the biogeochemical model as the one used to generated the "abio" fields ? In other words: do differences between those two simulations only reflect online vs offline effect or also a change in the particle concentrations / fluxes ? Do you expect an optimized coefficient change with consequent particles distribution changes (as possibly expected in paleo studies) ?"

The differences between "abio" and "bio" are stated in line 107-110, line 155-159, line 268-272. CESM have active marine ecosystem module, which can simulate particle fluxes online. The climatology under normal year forcing is shown in section 2.2. The exactly difference between eco and abio is that eco uses particle fluxes simultaneously computed by ecosystem, while abio uses particle fluxes fixed at prescribed values, which is the climatology of the same model (CESM) under the same forcing (normal year).  Eco and abio Nd can be turned on separately or together during the model set up steps.  So they can be run under the same model setup.  (All experiments in this study are under the same model setup: active ocean under normal year forcing).

[Figure]

The difference between climatology abio and climatology bio results is quit small if we compare the magnitudes in Figure 12 with the magnitudes in Figure 5, 9 and 10. This is intuitive as the physical circulation is the same, the particle fields are one with seasonality (eco) and another is climatology (abio). But if we run paleosimulations, the particle fields produced by the active ecosystem module will be different from the present day condition. In that case, we anticipate there will be much larger difference between abio and eco results.

We will apply this optimal parameter tuned for present day to paleo studies since we don't have a better option. First of all, our knowledge of past Nd sources and $\varepsilon$Nd are limited and paleo observation is much less compared with present day. It is not practical to tune the model parameters under paleo condition (for example, Last Glacial Maximum). Secondly, as we show in Table 2, EXP1 and EXP2 are slightly different parameters from CTRL, but the results are similar. Model sensitivity is quite small around the parameter setting in CTRL (Fig.3). And the particle fields in our model tuning process is different from Rempfer et al. 2011, although the general patterns are the same, we get similar optimal parameters. Thirdly, as shown in Figure 14 and Figure 16, we double or half those two parameters, but changes in $\varepsilon$Nd are very small. $\varepsilon$Nd is the proxy widely used in paleo studies. Therefore, we feel justified to use the parameters in paleo studies. Also, as Rempfer et al., 2012 pointed out, substantial changes in $\varepsilon$Nd is smallNd sources are required to generate large-scale changes in deep water $\varepsilon$Nd comparable to deglacial $\varepsilon$Nd reconstructions. But this is definitely a drawback of the model and should be kept in mind.

"l397: can't this shift in depth be attributed to a too slugish AMOC that favors vertical cycling rather than lateral advection ? Actually, the core of NADW visible from eNd data is rather 1500m than 3000m used here. l404-406: it looks that you use the same justification as Rempfer et al. (2011). What could be the drawbacks of the "sources simplifications" ? As you have a higher resolution of the CESM model available, did you test if improving the resolution helps reducing your biases ?"

[Figure]

The AMOC strength is our simulation is about 16.6 Sv, it is in the range of present day estimation. Therefore, AMOC is not sluggish in our model. Profiles in North and South Atlantic (Figure 10, profile 5 and 7), the less radiogenic NADW core is around 3,000 m.

Figure 10 profile 4 show model-data mismatch: observation show a very large vertical gradient for over 5 $\varepsilon$Nd unit within 3,000m, while model show a much less vertical gradient and more homogenous $\varepsilon$Nd at this location. This is deep convection region, therefore, the watermass should be well mixed, as shown in the [Nd]d profile in Figure 9 profile 4. Probably the very unradiogenic value near the surface in the observation have some surface input which our model don't have. It is impossible for the model to simulate every point consistent with observation, especially considering our simplifications on Nd sources. For example, the $\varepsilon$Nd field prescribed for dust source (Figure 2c) only have a large scale gradient and a homogeneous value in each basin; We use a global uniform boundary source magnitude (fboundary). Our goal to capture the general picture of the distribution of [Nd]d and $\varepsilon$Nd.

Our model resolution is improved from Rempfer et al. 2011, but our 3° model is still rather coarse. We haven't tried a higher resolution model. We can take a look if this is improved when our 1 degree run with all other traces is ready, but we don't think it will make a large difference. Our $\varepsilon$Nd fields prescribed for the boundary, river source (Jeandel et al., 2007) and dust source (Tachikawa et al., 2003) is also very low resolution. Therefore, if we use a high resolution model but with low resolution sources, we don't think it will help a lot. But if in the future, we have much higher resolution observations about Nd sources, it will definitely help to improve regional model-data inconsistency. Again, our point is that our model is able to capture the big picture and resolve small scale regional model-data consistency is out of the scope of this study. It may be improved in the future by using some regional model, or using a ensemble Kalman filter to have better parameterization in different regions(Liu et al., 2014).

"l425: although this sensitivity test has been performed in Rempfer et al. (2011), you should precise what your motivations are for performing such a sensitivity test."

These sensitivity tests have been run during the parameter tuning process. Here we show these results to confirm we have the same sensitivity as in Rempfer et al. 2011.

"Fig2: it does seem that you include oceanic ridges in your sources. Wether ridges are sources (in the Pacific ?) or sinks (as we've thought for a long time), it is very unlikely that values can be -10/ 5 eNd."

We follow the method in Rempfer et al. 2011 by applying a boundary source at the continental margins above 3,000 m. Figure 2b show the grids above 3,000 m in our model. The $\varepsilon$Nd prescribed for the boundary source is extrapolated from (Jeandel et al., 2007) for a global coverage. The boundary source is 3.57*10-5 g/(m2s) (fboundary/Total_area). The average sink in this bridge in Pacific is 4.2*10-12 g/(m2s) (Nd_p*w*thickness of the grid). Therefore, it is a local source in our model.

"Fig8: should put a colorbar here"

Thanks for suggesting this. Putting a colorbar will help the readers. Colors here refer to different depth range.

"Fig12: if you only look at online vs offline effects here (not sure this is the case), we would rather expect to see variations near the surface, which should be more relevant to look at."

Figure b and d shows the vertical difference. The maximum differences between "abio" and "eco" are near the surface, but the magnitude of the maximum differences is still very small compare with the magnitude in Figure 5.

Reference: Huang, K.-F., Oppo, D. W. and Curry, W. B.: Decreased influence of Antarctic intermediate water in the tropical Atlantic during North Atlantic cold events, Earth Planet. Sci. Lett., 389, 200–208, doi:10.1016/j.epsl.2013.12.037, 2014. Jahn, A., Lindsay, K., Giraud, X., Gruber, N., Otto-Bliesner, B. L., Liu, Z. and Brady, E. C.: Carbon isotopes in the ocean model of the Community Earth System Model (CESM1), Geosci. Model Dev., 8(8), 2419–2434, doi:10.5194/gmd-8-2419-2015, 2015.

[Figure]

Jeandel, C., Arsouze, T., Lacan, F., Techine, P. and Dutay, J.: Isotopic Nd compositions and concentrations of the lithogenic inputs into the ocean: A compilation, with an emphasis on the margins, Chem. Geol., 239(1-2), 156–164, doi:10.1016/j.chemgeo.2006.11.013, 2007.

Large, W. G. and Yeager, S. G.: The global climatology of an interannually varying air–sea flux data set, Clim. Dyn., 33(2-3), 341–364, doi:10.1007/s00382-008-0441-3, 2008.

Liu, Y., Liu, Z., Zhang, S., Jacob, R., Lu, F., Rong, X. and Wu, S.: Ensemble-based parameter estimation in a coupled general circulation model, J. Clim., 27(18), 7151–7162, doi:10.1175/JCLI-D-13-00406.1, 2014.

Pahnke, K., Goldstein, S. L. and Hemming, S. R.: Abrupt changes in Antarctic Intermediate Water circulation over the past, Nat. Geosci., 1(12), 870–874, doi:10.1038/ngeo360, 2008.

Rempfer, J., Stocker, T. F., Joos, F., Dutay, J.-C. and Siddall, M.: Modelling Nd-isotopes with a coarse resolution ocean circulation model: Sensitivities to model parameters and source/sink distributions, Geochim. Cosmochim. Acta, 75(20), 5927–5950, doi:10.1016/j.gca.2011.07.044, 2011.

Rempfer, J., Stocker, T. F., Joos, F. and Dutay, J.-C.: Sensitivity of Nd isotopic composition in seawater to changes in Nd sources and paleoceanographic implications, J. Geophys. Res., 117(C12), C12010, doi:10.1029/2012JC008161, 2012. Tachikawa, K., Athias, V. and Jeandel, C.: Neodymium budget in the modern ocean and paleo-oceanographic implications, J. Geophys. Res., 108(C8), 3254, doi:10.1029/1999JC000285, 2003.

Xie, R. C., Marcantonio, F. and Schmidt, M. W.: Deglacial variability of Antarctic Intermediate Water penetration into the North Atlantic from authigenic neodymium isotope ratios, Paleoceanography, 27(3), doi:10.1029/2012PA002337, 2012.

---

## Author Comment (AC2) · 21 Apr 2017

First of all, we thank the reviewer for his/her time for constructing the comments. We should have emphasized that the main objective of this paper is to document the details of Neodymium isotopes in the CESM. Indeed, it is a follow-up of Jahn et al. 2015, which describes carbon isotope implementation in the CESM. We are still working on implementing more different isotopes in the CESM for the purpose of a capability of a direct model-data comparison, which will help us to better understand past climate changes in terms of better interpretations of different proxy records as well as model validation. Since CESM is a community model and Nd module will be included in the next release of CESM, which will be used by many users, we feel it is necessary to document Nd implementation in the CESM. This is a technical paper, which describes and documents a new feature of the CESM, therefore, we submit it to GMD. Improving

Nd cycle is not the purpose of this paper, but probably will be done in the future work.

Although we have largely followed the method in Rempfer et al. 2011, there are several new features in our work. Firstly, we are using a more sophisticated model and it is not obvious that two totally different model will produce similar results. Therefore, it confirms the robustness of their method. Secondly, we are tuning parameters for the CESM using a more completed data, but it gives similar parameters as in Rempfer et al. 2011, which confirms the robustness of the overall magnitude of the parameters. Thirdly, we have eco Nd, which is coupled with active marine ecosystem, as well as abio Nd, which is similar as Rempfer et al. 2011. The small difference between eco nd and abio Nd is anticipated because the particle fields used in abiotic Nd module is the climatology of the particle fields in the marine ecosystem. Therefore, in principle, when equilibrium is reached, abiotic and biotic should produce the same climatological Nd fields (there are some seasonal variations, but the comparison showed in the paper are all decadal mean). However, if we use this module to do paleo simulation, in which both circulation and particle fields are changing, these two versions of Nd have the advantage that we can separate the two effects. But this kind of simulation is out of the scope of this study and we are working on some paleo simulations and will address this abiotic and biotic difference in our future work.

In the following, we have addressed all comments, with the original review text quoted. "(1) Way to optimize tuning parameters and evaluate simulation performance fboundary and [Nd]p/[Nd]d are optimized by cost function J (Figure 3). There is no information about spatial distribution of difference between observation and model simulation except for several selected profiles (Figures 9 and 10). Histograms (Figures 6 and 7) only present the trend in major oceanic basins for four depth layers. The size and spatial distribution of difference between observation and model will provide the information about under and/or over-estimation of source and sink terms. For instance, I would like to see the results of the tuning that is realised separately for different oceanic basins (Atlantic and IndoPacific separately and Southern Ocean as buffer zone to ensure the
continuity, for example). The upper layers affected by dust and river water will be treated separately from and the lower layers. With higher spatial resolution and more observational data relative to the previous study, such treatment would be possible. Since lithology and distance from continental margins are different between Atlantic and IndoPacific, it is not surprising that different parameterization lead to better simulation of seawater Nd concentration and "Nd values. About the evaluation of simulation performance, the authors continued to use a track of vertical sections from Atlantic to Pacific (Figure 2a). Because of large gradient of Nd concentration and "Nd from Atlantic to Pacific, moderate amplitude of discrepancy is not visible with this presentation. Basin-scale transect is more appropriate for this study. About the criteria of good agreement (3 "-units) should be revised because this size is equivalent or larger than changes in glacial/interglacial intermediate/deepwater "Nd values."

In addition to the several selected profiles (Figure 9 and 10), histograms (Figure 6 and 7), the spatial distribution of difference between observation and model simulation can also be seen in Figure 5 (the transect from the North Atlantic to the North Pacific). Figure 8, the scatter diagram, is another more direct comparison. We are trying to show our model is able to capture the general features of both [Nd]d and  $\varepsilon$ Nd.

Thanks for suggesting tuning parameters in different regions. It is a very good idea. Indeed, we are planning to use an ensemble Kalman filter method to perform parameter estimation for this ocean model and it will be interesting to optimize the parameters in different regions (Liu et al., 2014) in the future.

We agree that for the transect, a basin scale plot is more appropriate. Please see this in the Figure attached (Fig.1 in this AC).

This +/- 3  $\varepsilon$ Nd for validating is from Rempfer et al., 2011, we use their measurement as a benchmark, therefore we can compare with their results quantitatively.

"2) Assumption of homogeneous Nd flux from margins This assumption was already questioned in the study of Rempfer et al. (2011) by the authors themselves ("a globally

GMDD
uniform flux of fbs probably is not valid"). It will be really interesting to tackle this difficult issue because there are some new evidences. The first clue is the partial dissolution of river particle. This potential source had been considered independently from margins before the idea of the boundary source is generally accepted. A recent study on Amazon river mouth demonstrates the dissolution of detrital fraction and Nd release to the ocean (Rousseau et al., 2015). Since river runoff was simulated in CESM, river sediment flux could be quantitatively evaluated by assuming ratio(s) between dissolved and solid phases, a partial dissolution rate and a Nd concentration in solid phase. It is a similar treatment to dust Nd flux. This consideration will contribute to establishing weighted Nd flux from margins. The second clue is Nd release from poorly chemically weathered detrital fraction in relation to the dynamics of cryosphere (Howe et al., 2016). Howe et al. (2016) indicated detrital Nd contribution in the Labrador Sea due to Laurentide ice sheet retreat in the early Holocene. At present, glacier and ice sheet retreat at high latitudes during warm seasons could form Nd flux to the ocean by similar processes. Even if it will be difficult to quantitatively estimate such Nd flux, some sensitivity tests will provide new insight into Nd flux from this source."

Thanks for the references and suggestions. A more realistic boundary source will definitely improve model performance and also improve our understanding of Nd cycle. With more and more available observations, it is doable, for example, using the parameter estimation by data assimilation mentioned above to estimate the boundary source in different regions. But it is out the scope of our current work.

"Specific or minor comments Figures 9 and 10: What are the criteria of selection to show the profiles comparing Nd and "Nd values between observation and simulation?"

We pick these profiles as in Rempfer et al. 2011 to have a inter comparison.

"More recent compilation of seawater Nd and "Nd as well as Holocene "Nd values of sedimentary authigenic fraction and biogenic carbonate by Tachikawa et al. (in press) provides hydrography parameters (temperature, salinity, nutrients) that could be useful
for data model comparison, for instance with Figure 11."

Thanks for pointing out this new compilation of data. More available observation is always good. The first version of this study is using the data by Lacan et al., 2012. After van de Flierdt et al., 2016 (which is about double the amount of Lacan et al. 2012) is available, we re-do all the parameter tuning and analysis using more complete data set. However, there is no significant improvement as shown in the table attached (Fig2 in this AC). CTRL\_new is the optimal parameter using the new data (van de Flierdt et al., 2016). CTRL\_old is the optimal parameter using the old data (Lacan et al., 2012). CTRL\_R\_new is using parameters in Rempfer et al. 2011 and the new observation data. CTRL\_R\* is results in Rempfer et al. 2011 (old data).
Fig. 1. [Nd]d and eNd track in the Atlantic and Pacific basin
Interactive

comment

| Exp       | $[Nd]_p$ | f boundary | Inventory | $\tau_{Nd}$ | J [Nd]d | $J\epsilon_{Nd}$ | Jı  | J 2 |
|-----------|----------|-----------------------|-----------|-------------|--------------------|------------------|-----|----------------|
|           | $[Nd]_d$ | (g yr-1)              | (g)       | (yr)        | (pmol kg-1)        |                  | (%) | (%)            |
| CTRL_new  | 0.0009   | 4×109                 | 4.3×1012  | 785         | 8.1                | 1.76             | 72  | 82             |
| CTRL_old  | 0.0008   | 4×109                 | 5.0×1012  | 900         | 9.6                | 1.8              | 71  | 82             |
| CTRL_R_ne | 0.001    | 5.5×10 9   | 5.1×1012  | 720         | 9.3                | 1.78             | 64  | 83             |
| w         |          |                       |           |             |                    |                  |     |                |
| CTRL_R*   | 0.001    | 5.5×109               | 4.2×1012  | 700         | 9                  | 1.66             | 70  | 83             |

Fig. 2. Parameters and general performance of different experiments

---

## Editor Comment (EC1) · G. Munhoven (Editor) · 16 May 2017

**Referee comments**

During the public discussion of your manuscript two referee comments have been posted, to which you have replied in a particularly timely manner – thank you!

While both referees appreciate the thorough and rigorous validation of the model, they also emphasize the lack of originality in the developments reported. They find that the manuscript brings

• no significant improvement of the understanding of the Nd oceanic cycle;

- no improvement of Nd cycle modelling;
- no intercomparison of Nd modelling with different oceanic models.

As summarized by Referee #1, the current version of the paper merely reproduces with the CESM what Rempfer et al. (2011) have done before, including exactly the same sensitivity tests and diagnostics, and generally reaches similar conclusions.

Both referees nevertheless make constructive suggestions about how to adapt the manuscript in order to move it away from the level of a "technical internal note" (Referee #1) and to make it a valuable reference for future studies involving Nd and its isotopes with the CESM. Referee #1, e. g., explains that the submitted version is lacking

"[...] an accurate comparison between these new results and Rempfer et al. (2011) (or Arsouze et al. (2009)) in order to identify if some specific drawbacks of one model in some areas were not reproduced by another model, etc.";

Referee #2 suggests to focus on the parameter tuning procedure which could be

"[...] realized separately for different oceanic basins (Atlantic and Indo-Pacific separately and Southern Ocean as [a] buffer zone to ensure the continuity, for example). The upper layers affected by dust and river water will be treated separately from and the lower layers."

Referee #2 furthermore outlines ways to address the impact of the assumption of homogeneous margin Nd fluxes (which has been questioned in the earlier work of Rempfer et al. (2011), i. e., by several of the co-authors of this paper) and even provides references to relevant recent papers that could be used to define meaningful sensitivity tests to make progress on the problems related to these margin Nd fluxes.
Finally, Referee #2 singles out the biotic vs. abiotic module as the most original contribution of the reported development works, but finds that only a small part of the text is devoted to it.

I have now reread your paper in the light of the referees' comments and looked up the relevant literature cited in the manuscript and in the referees' comments. In conclusion, I concur with the referees. They have certainly provided critical, but above all, care-, thoughtful and constructive reviews, for which I thank them.

**Replies to referees**

I have also reread the replies to the Referees and I find that the referees' major concerns and suggestions are, unfortunately, not fully addressed. The reply to Referee #1 does not mention the advisable comparison of this model's results with results obtained from other models.

As stated in GMD's *Manuscript type* information page, "[d]evelopment and technical papers usually include a significant amount of evaluation against standard benchmarks, observations, and/or other model output as appropriate." Obviously, there are no benchmarks available in this case and the model-data comparison merely reproduces the results of the Bern3D implementation described by Rempfer et al. (2011). The implicit recommendation of Referee #1 seems thus rather natural: please present some in-depth comparison with the previous implementation (or with the results of Arsouze et al. (2009), if you prefer to compare with a model of similar complexity than CESM).

Most unfortunately though, the concrete and outstandingly constructive suggestions of Referee #2 are not considered at fair value: the first one (tuning procedure) is relegated
to "future work" while the second one (margin Nd fluxes) is declared "out of scope."

It should be possible to adapt the Nd cycle implementation in the model without too much hassle so that different parameter values can be used in different ocean basins and depth layers, as suggested by Referee #2, without having to call upon a Kalman filter method to tune these parameters. Designing one or two meaningful sensitivity tests for the margin Nd fluxes should also be feasible on the basis of the provided reference papers!

In the replies to referees, I read that the document "[...] is a technical paper, which describes and documents a new feature of the CESM, which fits the scope of GMD." It is certainly correct that this paper a priori fits into the scope of GMD. However, although GMD *Model description papers* are expected to put a stronger focus on technical details than papers in other, less model-centric, journals, GMD papers must not reduce to technical reference notes only. We clearly expect that papers "present significant advance" (see Editorial 1.1, section 2.3) in the scientific research area that they contribute to. Furthermore, this paper presents *incremental model development*: new functionality is added to an existing model. We explicitly encourage submission of such papers to GMD, but they "must include a tangible and potentially useful advance related to model development." (Editorial 1.1, Introduction). As it currently stands, the expected tangible advance is not sufficiently developed in this paper. I believe that target could nevertheless be easily reached if the referees' comments and suggestions were pursued.

**Decision**

Referee #1 states in his/her report that he/she cannot recommend the paper for publication in GMD; Referee #2 concludes that more efforts are required if the paper is meant to become a reference for future studies using Nd isotopes in the CESM, and Interactive comment

accordingly recommends a complete overhaul and resubmission. In their evaluation reports, the two **referees unanimously recommend to reject the manuscript** in its present form.

In its current version, **the manuscript cannot be accepted for publication**. The amount of work required to address the concerns expressed by the referees and to take their most important suggestions into account – as well as the referees' unanimous advice – precludes "major revision" at this stage, I am afraid.

I do, however – just like the referees – see good potential in the manuscript and I strongly recommend to **revise your paper** following the recommendations made by the two referees **and submit a completely new version**. Please provide readers with a more systematic intercomparison with another Nd enabled biogeochemical model (the Bern3D or another one) and amend the developments along the lines laid out by Referee #2.

Regarding the abiotic vs. biotic module, which has been singled out as the most original contribution in this paper, I would like to add the following suggestion: why not include a simple sensitivity test, with some prescribed (possibly hypothetical) change, such as a reduced AMOC or modified export production in order to illustrate the potential of the approach. I also find the denomination "abiotic" somewhat misleading, as the underlying particle flux distribution includes a biogenic contribution, although a fixed one. Please take also advantage of the other comments and recommendations provided by the referees.

I am looking forward to reading your fresh manuscript.
**References**

- Arsouze, T., Dutay, J.-C., Lacan, F., and Jeandel, C.: Reconstructing the Nd oceanic cycle using a coupled dynamical – biogeochemical model, Biogeosciences, 6, 2829–2846, doi:10.5194/bg-6-2829-2009, 2009.
- GMD Executive Editors: Editorial: The publication of geoscientific model developments v1.1, Geosci. Model Dev., 8, 3487–3495, doi:10.5194/gmd-8-3487-2015, available at http://www.geosci-model-dev.net/8/3487/2015/, 2015.
- Rempfer, J., Stocker, T. F., Joos, F., Dutay, J.-C., and Siddall, M.: Modelling Ndisotopes with a coarse resolution ocean circulation model : Sensitivities to model parameters and source/sink distributions, Geochim. Cosmochim. Ac., 75, 5927–5950, doi:10.1016/j.gca.2011.07.044, 2011.